# Changes in the Diversity and Composition of Gut Microbiota of Red-Crowned Cranes (*Grus japonensis*) after Avian Influenza Vaccine and Anthelmintic Treatment

**DOI:** 10.3390/ani12091183

**Published:** 2022-05-05

**Authors:** Xinyi Zhao, Wentao Ye, Wei Xu, Nan Xu, Jiajun Zheng, Rong Chen, Hongyi Liu

**Affiliations:** 1The Co-Innovation Center for Sustainable Forestry in Southern China, College of Biology and the Environment, Nanjing Forestry University, Nanjing 210037, China; zxy19981104@njfu.edu.cn (X.Z.); yewentao@njfu.edu.cn (W.Y.); xuwei2001@njfu.edu.cn (W.X.); 18871854847@sina.cn (N.X.); z2657244667@163.com (J.Z.); 2Nanjing Hongshan Forest Zoo, Nanjing 210028, China; haze876@163.com

**Keywords:** red-crowned crane, gut microbiota, diversity and composition, avian influenza vaccine, anthelmintic

## Abstract

**Simple Summary:**

Gut microbiota homeostasis is significant to host health and well-being. Treatment of red-crowned cranes with avian influenza vaccines and anthelmintics has played pivotal roles in therapeutic management in zoos. This study provides evidence concerning if and how the avian influenza vaccine and anthelmintic treatment impact the gut microbiota of red-crowned cranes. For the first time, we revealed that the gut microbiota of red-crowned cranes is resilient to the avian influenza vaccine and anthelmintic, which may be affected to disorder in the short term but can recover to homeostasis over time. In addition, it is proposed that more controlled experiments should be performed properly to contribute to animal disease control and optimal management in the zoo.

**Abstract:**

Gut microbiota homeostasis is important for host health and well-being; however, drugs may affect the composition and function of the gut microbiota. Red-crowned cranes are a vulnerable species. Treatment of red-crowned cranes with avian influenza vaccines and anthelmintics has played pivotal roles in therapeutic management in zoos. To investigate the changes in the diversity and composition of gut microbiota after the avian influenza vaccine and anthelmintic treatment, we used 16S rRNA sequencing to obtain and compare the bacterial community composition before and after the treatment. The alpha diversity of the gut microbiota of red-crowned cranes decreased on the day of the treatment and then fluctuated over time. The composition of gut microbiota tended to be similar in the short term after the treatment, as supported by the beta diversity hierarchical cluster analysis. Only 3, 8, and 72 operational taxonomic units (OTUs) of the three individuals were shared among the five groups before and after treatment. The relative abundance of Firmicutes significantly increased to 99.04% ± 0.28% on the day of the treatment, in which the relative abundance of *Lactobacillus* was 93.33% ± 5.85%. KEGG pathways analysis indicated that the main function of the gut microbiota is involved in metabolism, and the present study indicates that the gut microbiota of red-crowned cranes is resilient to the avian influenza vaccine and anthelmintic, even disordered in the short term, and could recover over time. More individual experimentation and functional potential in metabolism are needed in the future to support animal disease control and optimal management in the zoo.

## 1. Introduction

Gut microbiota is often considered essential for host health and well-being because its ecological stability ensures that beneficial symbionts and their associated functions are maintained over time [1,2]. Correspondingly, in the case of dysbiosis, disruption of the normal balance between the gut microbiota and host [3], the host is more susceptible to various diseases such as obesity [4], inflammatory bowel diseases [5,6], and cancer [7]. In general, although the composition and taxonomy of gut microbiota vary markedly across biotic (such as host age, host sex, and host disease) and abiotic (such as diet composition, geographical environment, and seasonal variation) factors [8,9], the gut microbiota can recover from external perturbations to adapt to changes and maintain stability [10]. However, the gut microbiota can be dramatically altered by short-term drug interventions. Microbial regulatory changes in drug metabolism and drug-mediated changes in intestinal microbiota may have beneficial or harmful effects on the host [11,12]. For example, the microbial composition of hens fed a diet supplemented with Astragalus, a common herbal medicine with anti-inflammatory, immunostimulatory, antioxidative, and antiviral activities, was greater than that of the control group, which could play a vital role in the modulation of fecal microbiota [13]. Many studies have shown that treatment with drugs can result in changes in the function and composition of the gut microbiota and even disrupt the normal population and function of bacteria [11,14,15,16,17]. Firmicutes were found to be significantly depleted (*p* < 0.05) in the day 7 and day 14 monensin/virginiamycin-treated groups and the day 14 monensin/tylosin-treated group of the chicken cecal microbiome, as compared to the control group for each respective timepoint [17]. Additionally, changes in the quantity and composition of the gut microbiota are affected by gastric acid, malnutrition, and parasitic infections [18,19,20]. A study on Langya hens showed that excessively high stocking density drove the abundance of bacteria and fungi connected with health problems, and the gut microecology gradually reached a mature and balanced status with age [21]. The alpha diversity of the gut microbiota in rabbits infected with nematodes and mice infected with flagellates decreased significantly during the acute stage of infection [22,23]. Therefore, the gut microbiota is dynamic and continually changes in composition to adapt to changes in the internal and external environments, which is closely related to the health and disease of organisms.

Since the development of veterinary medicine, avian influenza vaccines, and anthelmintic treatment have played pivotal roles in the management of avian diseases caused by viruses and parasites [24,25]. Vaccines are the cornerstone of preparing for and combating potential pandemics [26]. The use of anthelmintic treatment to control parasitic infections has long been standard practice. The need to respond quickly to potential influenza pandemics and parasite prevention in birds is necessary for zoos. The red-crowned crane (*Grus japonensis*) is one of the largest wading birds and is listed as vulnerable (VU) according to the IUCN Red List (https://www.iucnredlist.org/, accessed on 7 April 2022) in 2021. This species was also classified as a national first-grade protected animal in China (http://www.forestry.gov.cn/, accessed on 7 April 2022). Measures including the establishment of biosphere reserves and artificial breeding programs [27] have been developed to maintain populations of red-crowned cranes in China, Japan, South Korea, and other countries where red-crowned cranes have been kept in captivity in many zoos or nature reserves [28,29,30]. A previous study on red-crowned cranes found that both captivity and artificial breeding influenced the structure and diversity of the gut microbiota, potentially due to changes in diet, vaccination, antibiotics, and living conditions [29]. With the development of high-throughput sequencing technology, the characteristics of changes in gut microbiota induced by various factors have been further revealed [31,32,33,34]. However, how the avian influenza vaccine, anthelmintic treatment, and stress during these processes affect the diversity, composition, and function of the gut microbiota of red-crowned cranes kept in captivity in the zoo needs to be further investigated.

Considering the above, and with little available information on the animals’ disease control and optimal management in the zoo, this study aimed to provide an elementary comparison and assessment of gut microbiota of red-crowned cranes before and after the avian influenza vaccine and anthelmintic treatment by high-throughput sequencing of the V3-V4 region of the 16S rRNA gene. Moreover, the effects of the avian influenza vaccine and anthelmintic treatment on the structure and function of the gut microbiota of red-crowned cranes were anticipated to be evaluated.

## 2. Materials and Methods

### 2.1. Animals

This study was approved by the Laboratory Animal Welfare and Ethics Committee of Nanjing Forestry University. Three healthy red-crowned cranes, two adults, and one subadult (Sample 1: S1; Sample 2: S2; Sample 3: S3), kept in Nanjing Hongshan Forest Zoo (32°11 N; 118°83 E) in Jiangsu Province, China, were enrolled in this study. They were subjected to routine physical examinations and disease prevention measures during the autumn. Albendazole (0.2 g/tablet, Sino-American Tianjin Smithkline Pharmaceutical, Tianjin, China), a high-efficiency and low-toxicity broad-spectrum anthelmintic drug, was used for deworming. Reassortant avian influenza virus (H5 + H7) trivalent vaccine (Zhaoqing Dahuanong Biological Pharmaceutical, Zhaoqing, China) was used to prevent avian influenza caused by the H5 subtype (clade 2.3.4.4d and clade 2.3.2.1d) and H7 subtype avian influenza viruses. Each crane was administered one tablet and injected with a 1 mL vaccine. Before that, they had been in good health and had received no other drugs or treatments in the past 6 months.

### 2.2. Sample Collection, DNA Extraction, and PCR Amplification

Fecal samples were collected before (10 d: B) and after (0 d: D0; 5 d: D5; 10 d: D10; 15 d: D15) the avian influenza vaccine and anthelmintic treatment. All samples were snap-frozen in liquid nitrogen and stored at −80 °C in the laboratory. Total DNA was extracted from the samples using an OMEGA Soil DNA Kit (D5625-01) (Omega Bio-Tek, Norcross, GA, USA). DNA concentrations were assessed using a UV spectrophotometer (NC2000, Thermo Scientific, Shanghai, China), and purity was monitored using 0.8% agarose gel electrophoresis.

DNA was diluted to 10 mM for PCR amplification of the V3-V4 region of the 16S rRNA gene using the specific primers F: ACTCCTACGGGAGGCAGCA and R: GGACTACHVGGGTWTCTAAT. For a final volume of 25 μL, 2 μL of template DNA was added to a solution containing 8.75 μL of water, 0.25 μL of Q5 high-fidelity DNA polymerase, 5 μL 5× reaction buffer, 5 μL 5× high GC buffer, 2 μL dNTP (10 mM), and 1 μL of each primer (10 μM). The reaction conditions were as follows: initial denaturation at 98 °C for 30 s, followed by 27 cycles of denaturation at 98 °C for 15 s, annealing at 50 °C for 30 s, elongation at 72 °C for 30 s, and finally, 72 °C for 5 min. Amplification was subjected to 2% agarose gel electrophoresis, and the target fragments were cut and recovered using the Axygen gel recovery kit.

### 2.3. Gene Library Construction and Sequencing

Sequencing libraries were constructed using the TruSeq Nano DNA LT Library Prep Kit (Illumina, NEB, Ipswich, MA, USA) following the manufacturer’s recommendations. Library quality was assessed by Agilent High Sensitivity DNA Kit using Agilent Bioanalyzer and quantified on the Promega QuantiFluor Quant-iT PicoGreen dsDNA Assay Kit (Invitrogen, Carlsbad, CA, USA). Paired-end reads sequencing at 2× 250 bp was then performed using the NovaSeq 6000 SP Reagent Kit (500 cycles) using Illumina NovaSeq at Shanghai Personal Biotechnology Co., Ltd. (Shanghai, China). 

### 2.4. Analysis of Sequences

All data analyses were performed at the Genescloud Platform (Shanghai Personal Biotechnology Co., Ltd., Shanghai, China) (https://www.genescloud.cn/home, accessed on 7 April 2022), using the R Programming Language (version 4.1.3) and QIIME2 (2019.4) software.

Sequence denoising or clustering was performed using QIIME2. The primer fragments of the sequence were excised with the qiime cutadapt trim-paired, and the unmatched primer sequences were discarded. DADA2 [35] was used for quality control, denoising, splicing, and de-chimerization using qiime dada2 denoise-paired. The remaining high-quality sequences were clustered into operational taxonomic units (OTUs) at 97% sequence identity, with a representative sequence selected from each OTU using default parameters [36]. OTU taxonomic classification was performed by blasting a representative sequence set from the Greengenes database. After obtaining representative OTU sequences, statistics on the length distribution of high-quality sequences in all samples were performed. OTU leveling was performed using rarefaction. A certain number of sequences were randomly selected from each sample to reach a uniform depth to predict the observed OTUs and their relative abundances in each sample at this sequencing depth [37,38]. 

To calculate alpha diversity, we rarified the OTU table and calculated two metrics: the Chao1 index [39] to estimate species richness and the Simpson index [40] to describe community diversity. Additionally, we used non-metric multidimensional scaling (NMDS) analysis and hierarchical clustering in beta diversity analysis. The NMDS analysis is not affected by the numerical value of the sample distance, and only the size relationship between them is considered. For data with complex structures, the sorting results may be more stable. The Bray–Curtis distance matrix was used for NMDS analysis. Clustering analysis displayed the similarity between samples in the form of a hierarchical tree and measured the quality of the clustering effect based on the branch length of the clustering tree. The Bray–Curtis distance matrix adopts the UPGMA algorithm (average) for cluster analysis.

Species and community composition analyses are represented by Venn diagrams and bar plots. Venn diagrams were used to count the number of species according to the uniqueness of each group and OTUs shared between the groups. According to the bar plots of the community, two aspects of the results are visualized intuitively: (i) what specific composition of microbial communities are contained in each sample at the taxonomic level (phylum and genus); (ii) the relative abundance of microbial phyla and genera in the fecal samples of red-crowned cranes are shown visually to understand the composition of the community structure of different samples at each taxonomic level. The relative abundance is represented as the mean ± standard deviation.

Phylogenetic Investigation of Communities by Reconstruction of Unobserved States (PICRUSt2) [41] was used to predict the functional abundance of samples and explore functional profiles [42]. PICRUSt2 can predict 16S rRNA gene sequences in the Kyoto Encyclopedia of Genes and Genomes (KEGG) pathway database, in which metabolic pathways are classified into six categories: metabolism, genetic information processing, environmental information processing, cellular processes, organismal systems, and human diseases. We used the normalized pathway abundance table to calculate the average abundance or the total number of primary and secondary pathways depending on the selected samples.

## 3. Results

### 3.1. DNA Sequence Data

A total of 956,460 raw paired-end reads were obtained from the Illumina HiSeq of 16S rRNA gene amplicons from 15 samples. A total of 639,123 (66.82%) high-quality reads remained for analysis after chimeras were filtered out and low-quality sequences were removed, with an average read length of 424 bp, ranging from 25,685 to 73,802 reads per sample. After subsampling reads to the same sequencing depth (34,558 sequences) between samples, 13855, OTUs clustered at 97% sequence identity (Appendix A).

### 3.2. Alpha Diversity and Beta Diversity Analyses

The alpha diversity of the samples at the OTU level was estimated using the Chao1 and Simpson indices (Figure 1). Both the Chao1 and Simpson indices of S2 and S3 dropped on the day after the avian influenza vaccine and anthelmintic treatment (D0), whereas they fluctuated between 5 and 15 days. The alpha diversity of these samples suggested that the richness and community diversity of the gut microbiota of red-crowned cranes could be quickly influenced by avian influenza vaccine and anthelmintic treatment and gradually recovered over time. 

Beta diversity was focused on the comparison of diversity among different samples. From the NMDS results (Appendix A), the stress was 0.106 (stress < 0.2), indicating that the NMDS analysis could reliably reflect the degree of difference between the samples. According to beta diversity hierarchical cluster analysis (Figure 2), a multivariate statistical analysis method, samples in accordance with their degree of affinity in genera composition were classified, with an abundance ranking stacked histogram of the top 10 genera. The shorter the number of branches between the samples, the more similar the genus composition of the two samples. The bar plot shows that the genera composition of most samples in groups D0, D5, and D10 were similar, with the maximum abundance of *Lactobacillus*. Additionally, three samples in Group B clustered together, and two samples in Group D15 clustered together. These results showed that the gut microbiota of red-crowned cranes formed a steady state before the avian influenza vaccine and anthelmintic treatment. Fifteen days after the treatment, the gut microbiota recovered to another steady-state, which was different from that before the treatment.

### 3.3. Microbial Community Composition of Samples at Different Taxonomic Levels

We analyzed the community composition of the fecal samples of red-crowned cranes in different groups (B, D0, D5, D10, and D15) (Figure 3) at the phylum level (Figure 4a, Appendix A) and genus level (Figure 4b, Appendix A). The bar plots show the percentage of community abundance at different taxonomic levels. The dominant microbial phyla (top 10) and genera (top 15) are shown. As shown in Figure 4a, comparing the microbial phyla of samples in groups B and D0, although the composition of samples in group B was not similar, the relative abundance of Firmicutes significantly increased to 99.04% ± 0.28% in group D0, which meant that the bacteria of other phyla were instantly reduced after avian influenza vaccine and anthelmintic treatment. In groups D5, D10, and D15, Firmicutes was still the most abundant phylum, accounting for 91.15% ± 3.82%; 66.39% ± 22.18%; 57.76% ± 16.67%, respectively, whereas the relative abundance of Proteobacteria and Actinobacteria also increased. At the genus level (Figure 4b), in group D0, the relative abundance of *Lactobacillus* was 93.33% ± 5.85%. The relative abundance of *Lactobacillus* showed a decreasing trend in the samples after the treatment, whereas the other microbial genera in S1 and S2 increased. However, S3 was a subadult, younger than S1 and S2, which may lead to dissimilar disorder and recovery state and ecological stability of microbial communities affected by the treatment.

### 3.4. Functional Gene Analysis

Six primary metabolic pathways and secondary metabolic pathways (top three) of all fecal samples of red-crowned cranes were annotated using KEGG pathways. The maximum percentage of primary metabolic pathways (78.92%) was found to be involved in metabolism, in which carbohydrate metabolism (Figure 5A), amino acid metabolism (Figure 5B), and the metabolism of cofactors and vitamins (Figure 5C) were the most abundant secondary metabolic pathways. Furthermore, 14.40% of the primary metabolic pathways were involved in genetic information processing, with the top three being replication and repair (Figure 5L); translation (Figure 5M); and folding, sorting, and degradation (Figure 5N).

## 4. Discussion

The present study employed high-throughput sequencing of the V3-V4 region of the 16S rRNA gene to explore the effects of the avian influenza vaccine and anthelmintic treatment on the gut microbiota of red-crowned cranes. We aimed to evaluate whether and how the treatment might lead to changes in the diversity and composition of gut microbiota. The results showed that alpha diversity indices were reduced by the avian influenza vaccine and anthelmintic treatment, even though the reduction was limited. Additionally, there were few common OTUs in the five groups, indicating that the composition of the gut microbiota of red-crowned cranes was greatly affected. The effects of antibiotics and anthelmintics on the alpha diversity of other animals are similar to our results [9,43]. Moreover, the gut microbiota of the recovery group of brown frogs after 7 days was not significantly different from that of the gentamicin group [43], which also proved that alpha diversity gradually recovered over time. High bacterial diversity benefits the overall animal health and productivity [44]. The use of drugs reduces the alpha diversity of the gut bacterial community, which may be a manifestation of either the adverse side effects of drugs or disorders of the gut microbiota [45]. However, the beta diversity hierarchical cluster analysis of our study demonstrated that the gut microbiota of red-crowned cranes was resilient to the treatment in the short term. The organism has a certain ability to reshape gut microbiota homeostasis, but the microbial compositions of the samples in group D15 were different from the compositions of the samples in group B. So, further studies are needed to verify whether the microbial composition is likely to finally recover to the composition similar to that before the treatment. It is significant to study its dynamic ecological stability over prolonged periods [46].

According to the results of microbial community composition of samples at the phylum and genus levels, we found that Firmicutes had a high proportion in almost all samples, with the highest abundance of *Lactobacillus* in group D0. Interestingly, some gut anti-microbial proteins may affect the composition of the gut microbiota, typically increasing the proportion of *Lactobacillus*, demonstrating that increased *Lactobacillus* may promote gut mucus-layer homeostasis [47]. Additionally, *Lactobacillus* can strengthen epithelial defense functions, modulate the gut microbiota, and maintain the gut barrier [47,48], which is similar to the findings of our study. Thus, *Lactobacillus* may help maintain gut microbiota stability. Previous studies have shown that the gut microbiota of wild, captive adolescent, captive adult, artificially bred adolescent and artificially bred adult cranes are dominated by three phyla: Firmicutes (62.9%), Proteobacteria (29.9%), and Fusobacteria (9.6%) [29]. However, Fusobacteria existed in S1 and S2 before the avian influenza vaccine and anthelmintic treatment, and the relative abundance of Fusobacteria was quite low after the treatment in our study. We may infer that the treatment influenced this phylum. In contrast, the relative abundance of Fusobacteria was low in all S3 samples. Whether the subadult, age, and dietary preferences have such effects on the Fusobacteria of intestinal bacteria requires further investigation.

Immunity develops approximately 14 days after the reassortant avian influenza virus (H5 + H7) trivalent vaccine, and the immune period of chickens is approximately 6 months [49]. Albendazole is a benzimidazole derivative with an efficacy profile [50]. Approximately 2–2.5 h after oral administration, the blood drug concentration of animals reaches a peak. Albendazole and its metabolites (including albendazole sulfoxide and albendazole sulfone) were discharged from the urine and feces within 24 h, without accumulation in the body [50,51]. We collected fecal samples of red-crowned cranes after 3 h on the day after the avian influenza vaccine and anthelmintic treatment (D0). Therefore, we inferred that the effects on the gut microbiome were largely driven by anthelmintics instead of vaccines. However, due to the limited number of red-crowned cranes in the Nanjing Hongshan Forest Zoo, our inference deserves more individual experimentation and consideration, and functional potential in metabolism requires further metagenomic sequencing.

Furthermore, gut bacterial homeostasis appeared to recover with a daily diet over time in our study. The gut microbiome can be affected by diet, and different dietary interventions are used by poultry producers or zookeepers to enhance avian growth and reduce the risk of enteric infection by pathogens [52]. A comprehensive understanding of the effects of disease control measures or treatments, diets, and stress will help develop new dietary or management interventions to promote the growth of red-crowned cranes, maximize the use of host feed, and protect birds from intestinal diseases caused by avian influenza, parasites, and pathogenic bacteria.

## 5. Conclusions

Overall, the present research provides evidence concerning if and how the avian influenza vaccine and anthelmintic treatment impact the gut microbiota of red-crowned cranes. The results suggest that the gut microbiota of red-crowned cranes is resilient to avian influenza vaccine and anthelmintic, which may be affected to disorder in the short term but can recover to homeostasis over time. Additionally, more controlled experiments should be performed properly to understand the effects on the gut microbiome driven by anthelmintics and vaccines, respectively. Finally, since the gut microbiota is a very dynamic ecosystem, we hope to make a contribution to understanding the factors leading to the microbiota dysbiosis and the relationship between species forming gut microbiota.

## Figures and Tables

**Figure 1 animals-12-01183-f001:**
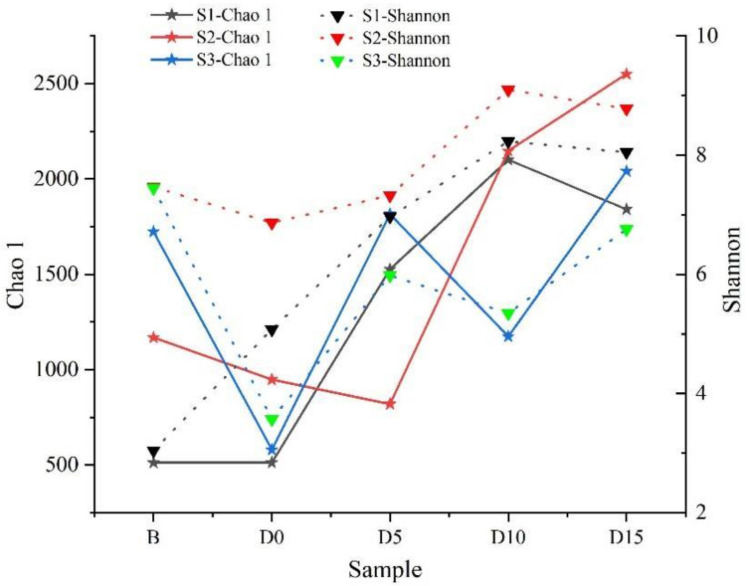
Alpha diversity of the intestinal bacterial community of the samples of red-crowned cranes. Comparison of the alpha diversity (Chao1 and Shannon) of the intestinal bacterial community of red-crowned cranes before and after the avian influenza vaccine and anthelmintic treatment.

**Figure 2 animals-12-01183-f002:**
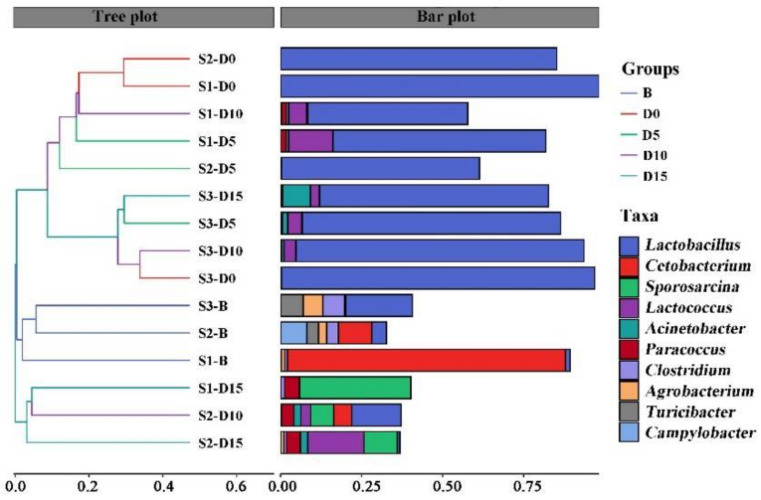
Beta diversity hierarchical cluster analysis of microbial genera of the fecal samples in different groups (B, D0, D5, D10, D15) of red-crowned cranes. The panel on the left is a hierarchical clustering tree diagram, in which samples are clustered according to their similarity with each other. The panel on the right is the abundance ranking Stacked histogram of the top 10 genera.

**Figure 3 animals-12-01183-f003:**
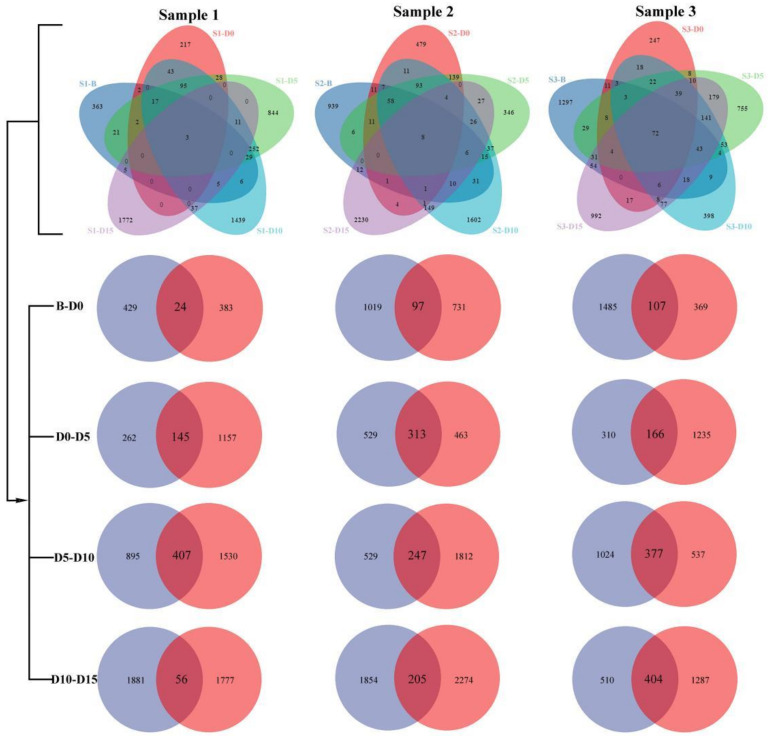
Venn diagrams. Venn diagrams show the numbers of OTUs (97% sequence identity) that were shared or not shared of the fecal samples of each individual (S1, S2, and S3) and shared OTUs between different groups (B and D0; D0 and D5; D5 and D10; D10 and D15).

**Figure 4 animals-12-01183-f004:**
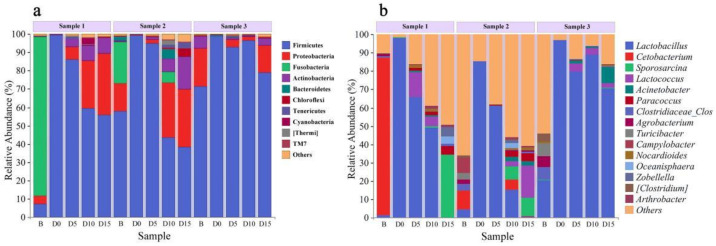
Differences in relative abundance of microbial phyla and genera of the fecal samples of red-crowned cranes in different groups (B, D0, D5, D10, D15). Differences in abundance of dominant microbial phyla (**a**: top 10 are shown) and genera (**b**: top 15 are shown) of five groups (B, D0, D5, D10, D15) of each individual (S1, S2, and S3).

**Figure 5 animals-12-01183-f005:**
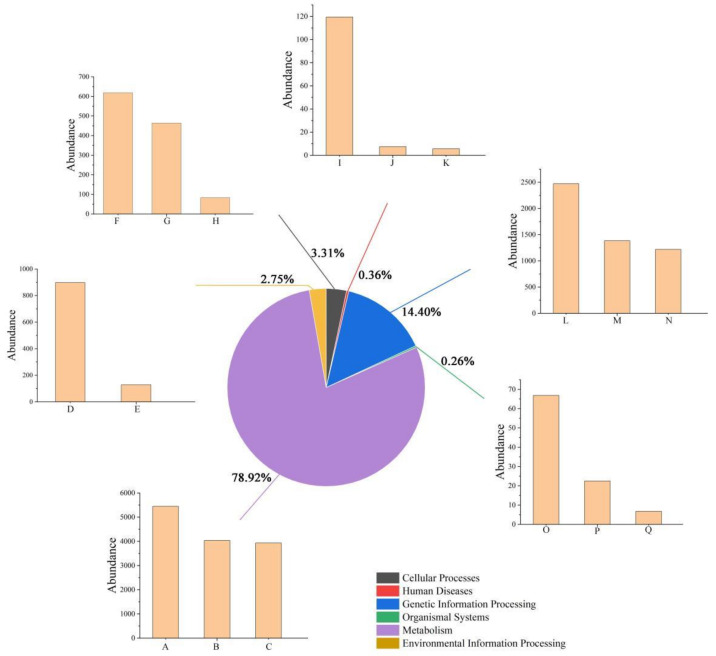
Distributions of primary and secondary metabolic pathways of all fecal samples of red-crowned cranes annotated by KEGG pathways. Pie chart represents primary metabolic pathways and numbers indicate relative % representation. Bar charts represent secondary metabolic pathways (top 3 are shown). A: Carbohydrate metabolism; B: Amino acid metabolism; C: Metabolism of cofactors and vitamins; D: Signal transduction; E: Membrane transport; F: Cell growth and death; G: Cell motility; H: Cellular community prokaryotes; I: Infectious diseases; J: Neurodegenerative diseases; K: Cancers; L: Replication and repair; M: Translation; N: Folding, sorting, and degradation; O: Environmental adaptation; P: Endocrine system; Q: Immune system.

## Data Availability

The nucleotide sequence data reported are available in the GenBank databases under the accession number PRJNA823535.

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
