# Peer review of "Changes in the Diversity and Composition of Gut Microbiota of Red-Crowned Cranes (Grus japonensis) after Avian Influenza Vaccine and Anthelmintic Treatment"

_animals, 2022, doi:10.3390/ani12091183_

Round 1
Reviewer 1 Report
Dear authors,
Thank you for the interesting article. Even though the number of birds is small, research is significant and further research would be very beneficial.
There are some minor concerns that I have incorporated into the text.
Also, it is necessary to provide a conclusion that will be supported by the obtained results of your research.
Additionally, English language grammar and style require minor editing and corrections.
All the best and stay safe

Reviewer 2 Report
- Abstract: it is not clear what the authors want to stated with the sentence “The maximum percentage of primary metabolic pathways (78.92%) was found to be involved in metabolism”. The sentence seems redundant. Please improve it.
- Line 44: authors should avoid to use the old term “microflora” and use microbiota or microbiome when appropriate.
- Lines 50 to 53 and 57 to 61. It should be better if the authors can give examples and focus on microbiota alterations in birds instead of pigs, rabbits, macaques and frogs.
- Authors need to order the figures in the manuscript and put them near the text when they are first mentioned.
- Authors don’t need to make references to the figures again in the discussion. Figures were already analyzed in the results sections. In addition, authors need to avoid a wide repetition of the results in the discussion.
- Authors stated in the results section “These results showed that the gut microbiota of red-crowned cranes formed a steady state before the avian influenza vaccine and anthelmintic treatment. Fifteen days after the treatment, the gut microbiota recovered to another steady-state, which was different from that before the treatment”. However, this point has not been deeply/clearly tackled in the discussion.
- I agree with the authors that the changes in the gut microbiome were probably driven by anthelmintic instead of vaccine. In this section authors propose “more individual experimentation”. Thus, it would be of value that they provide some references (if they exist) in with vaccinations of birds actually altered the gut microbiota.
Reviewer 3 Report
The manuscript entitled “Changes in the diversity and composition of gut microbiota of red-crowned cranes (Grus japonensis) after avian influenza vaccine and anthelmintic treatment” focuses on the very interesting and important topic – the gut microbiota. We already know that gut microbiota is considered a biological barrier protecting the gastrointestinal tract against colonization with infectious agents. Also, the complexity of the microbiota can be affected by many factors leading to dysbiosis. Since microbiota is a very dynamic ecosystem, understanding the relationship between species forming gut microbiota is very important.
The Authors assessed the gut microbiota of red-crowned cranes and obtained interesting results. However, the presented data raise some questions. First, the Author do not discuss why the birds were treated with albendazole and vaccinated against avian influenza virus at the same time. This issue should be presented and discussed because it is very unclear which factor is responsible for the alteration the complexity of microbiota. The low number of evaluated birds is understandable, but the whole experiment lacks proper controls. At least the Authors should assess the bird without any treatment or treated with albendazole. These topics should be discussed in the section discussion.
The Authors used the set dose of albendazole per bird. Should not the dose of albendazole be calculated per body weight, since in the studies participated two adult birds and one subadult? Please discuss this issue.
Minor remarks
Line 36: after “the host” verb “is” should be introduced
Line 37: “malnutrition” is not a disease, please change; as a result of dysbiosis the adsorption of nutrients is worse, and that could lead to malnutrition
Line 56: it should be explained “high density of what drove the abundance…”
All names of bacterial species, genera and orders should be italicized, please correct
After revision I recommend this manuscript for publication
